# Sequencing of a central nervous system tumor demonstrates cancer transmission in an organ transplant

Marie-Claude Gingras[1,2] , Aniko Sabo[1] , Maria Cardenas[1], Abbas Rana[3], Sadhna Dhingra[4], Qingchang Meng[1], Jianhong Hu[1], Donna M Muzny[1], Harshavardhan Doddapaneni[1], Lesette Perez[1], Viktoriya Korchina[1] , Caitlin Nessner[1], Xiuping Liu[1], Hsu Chao[1], John Goss[3], Richard A Gibbs[1] 

Four organ transplant recipients from an organ donor diagnosed with anaplastic pleomorphic xanthoastrocytoma developed fatal malignancies for which the origin could not be confirmed by standard methods. We identified the somatic mutational profiles of the neoplasms using next-generation sequencing technologies and tracked the relationship between the different samples. The data were consistent with the presence of an aggressive clonal entity in the donor and the subsequent proliferation of descendent tumors in each recipient. Deleterious mutations in *BRAF*, *PIK3CA*, *SDHC*, *DDR2*, and *FANCD2*, and a chromosomal deletion spanning the *CDKN2A/B* genes, were shared between the recipients' lesions. In addition to demonstrating that DNA sequencing tracked a donor/recipient cancer transmission, this study established that the genetic profile of a donor tumor and its potential aggressive phenotype could have been determined before transplantation was considered. As the genetic correlates of tumor invasion and metastases become better known, adding genetic profiling by DNA sequencing to the data considered for transplant safety should be considered.

## Introduction

Although there has been a steady increase over the last decade in organ donation from deceased donors, there is still a significant discrepancy between organ availability and demand. For example, 32,321 recipients received organs from 11,870 donors in 2019, but 11,702 patients were removed from the transplant list because they died (5,604) or became too ill to qualify for a transplant (6,098). As of 11 October 2020, there were 119,465 patients remaining on the waiting list (https://optn.transplant.hrsa.gov/data/view-data-reports/national-data/). For this reason, donors deceased after cancer of the central nervous system (CNS) are considered for donation, representing 0.3% of the donor pool.

Transplantation of organs from a donor with a known malignancy carries potential risks of cancer transmission to immunosuppressed recipients. These risks vary depending on factors such as the type of tumor, the history of the malignancy and the treatment received. Guidelines on the safety of organs for transplantation have therefore been issued, based on several published studies and reports from organ sharing registries of different countries (https://www.edqm.eu/en/guide-quality-and-safety-organs-transplantation).

In general, primary tumors from the CNS rarely disseminate outside the cranial cavity (incidence of 0.4–2.3% [Pasquier et al, 1980]) and those that metastasize are predominantly of aggressive histological types such as glioblastoma multiforme (GBM) and medulloblastoma (Cavaliere & Schiff, 2004). However, cases of extracranial metastasis from lower grade tumors to the lungs, pleura, cervical lymph nodes, bone, liver, and intra-thoracic and intra-abdominal lymph nodes have been reported (Liwnicz & Rubinstein, 1979; Pasquier et al, 1980; Hoffman & Duffner, 1985). Several risk factors could influence dissemination, such as ventriculo-peritoneal shunts and chemo or radiotherapy (Cavaliere & Schiff, 2004). Consequently, transmission through transplantation of an organ with undetected metastasis has been known to occur (Lefrancois et al, 1987; Ruiz et al, 1993; Colquhoun et al, 1994; Jonas et al, 1996; Bosmans et al, 1997; Frank et al, 1998; Detry et al, 2000; Buell et al, 2001; Armanios et al, 2004; Collignon et al, 2004; Morath et al, 2005; Kashyap et al, 2009; Ison & Nalesnik, 2011; Warrens et al, 2012; Zhao et al, 2012; Nauen & Li, 2014), and the risk from all histological types of CNS malignancy has been estimated to be 1.5% (Morse et al, 1990; Watson et al, 2010) (www.odt.nhs.uk/). In such cases, the origin of transmission is confirmed by histology and the analysis of markers previously detected in the donor lesion.

In the last decade, next-generation sequencing (NGS) has been extensively used to characterize several tumor types and identify genes driving tumor aggressiveness in studies such as The Cancer Genome Atlas Program (TCGA) and International Cancer Genome Consortium. With the accumulation of these data, the role of genes, and the impact of their mutations, deletions, and amplifications are

---

[1]Human Genome Sequencing Center, Department of Molecular and Human Genetics, Baylor College of Medicine, Houston, TX, USA   [2]Michael E. DeBakey Department of Surgery, Baylor College of Medicine, Houston, TX, USA   [3]Abdominal Transplant Center, Michael E. DeBakey Department of Surgery, Baylor College of Medicine, Houston, TX, USA   [4]Department of Pathology, Baylor College of Medicine, Houston, TX, USA

Correspondence: mgingras@bcm.edu

becoming more precisely defined. Present studies are integrating various data types and are focusing on molecular relationships across cancer types to explore clinical action ability in cancer treatment (Hoadley et al, 2018). However, the potential of NGS in establishing the safety of transplant from donor with CNS cancers has not yet been investigated.

We report a case of a young organ donor diagnosed with anaplastic pleiomorphic xanthoastrocytoma (PXA) and multiple organ recipients who developed aggressive neoplasms shortly after transplantation (Fig 1). The newly developed tumors had similar morphology but lacked glial and neural markers characterizing the donor lesion, raising doubt on their origin. This prompted the following questions: Can the origin of the recipients' lesions be unequivocally determined by identifying the mutational profile of the neoplasms using the NGS technology and comparative genomic analysis? Further, can this study encourage the utilization of NGS based methods to reduce the risk of cancer transmission, following transplantation?

# Results

## Patients

Clinical data variables including gender, age, clinical events, operative procedure, and survival time, and cytological and pathological finding are presented in Tables 1 and S1, respectively.

### Donor
The donor was a young male who at the age of 15 yr and a half had two massive spontaneous hemorrhages in a 2-mo period. No tumor was detected at the time of the first hemorrhage, but magnetic resonance imaging performed 47 d after the second hemorrhage showed a large underlying tumor. Surgery was performed to evacuate the hematoma and a fragment of the tumor was collected. 10 d later, a subtotal tumor resection was performed and diagnosed as anaplastic pleiomorphic xanthoastrocytoma (PXA). Cytology revealed epithelioid and spindle-shaped neoplastic cells with large, oval nuclei and glassy cytoplasm. A massive regrowth was once again resected 56 d later and the tumor classification was upgraded from a

Grade II to a Grade III progressive astrocytoma. The tumor tested positive for BRAF V600E mutation by immunohistochemistry. The patient was administered an oral chemotherapy regimen consisting of dabrafenib and trametinib (inhibitors of the associated enzyme B-Raf and the mitogen-activated protein kinase [MEK] pathway which plays a role in the regulation of cell growth), temozolomide (alkylating agent used as a second-line treatment for astrocytoma), and palliative radiation. The patient died 346 d after the first diagnosis of cancer was established. Computed tomography axial scans of the chest, abdomen, and pelvis without intravenous contrast detected no signs of malignancy. His liver, kidneys, pancreas, and lungs were donated to four recipients.

### Liver recipient (LR)
The liver recipient was a 30-yr-old female diagnosed with Budd-Chiari syndrome and with a pathogenic JAK2 V617F (c.G1849T) variant. A liver biopsy performed 56 d after transplant identified pleomorphic malignant cells with abundant eosinophilic cytoplasm and marked nuclear atypia including prominent nucleoli but negative for glial marker (Fig 2A–C). The recipient died 85 d after transplant. At autopsy, the liver showed several hepatic lesions (Fig 2D) described as necrotic malignant pleomorphic neoplasms and several peritoneal metastases were observed spreading beyond the transplanted organ.

### Kidney recipient (KR)
The KR was a 33-yr-old male with a history of lung adenocarcinoma and pulmonary metastatic disease. A graft nephrectomy was performed 3 mo after transplant because of the detection of malignancy. The neoplasm was characterized by cells with large vesicular nuclei with coarse chromatin and prominent nucleoli, abundant eosinophilic cytoplasm, and morphology that ranged from spindled to rounded and epithelioid. Progressive omental and mesenteric carcinomatosis was observed by computed tomography scan 15 d after the nephrectomy. The patient survived and was still alive 2 yr later.

### Kidney and pancreas recipient (KPR)
The kidney and pancreas recipient was a 55-yr-old male with a past medical history of end-stage renal disease due to type I diabetes

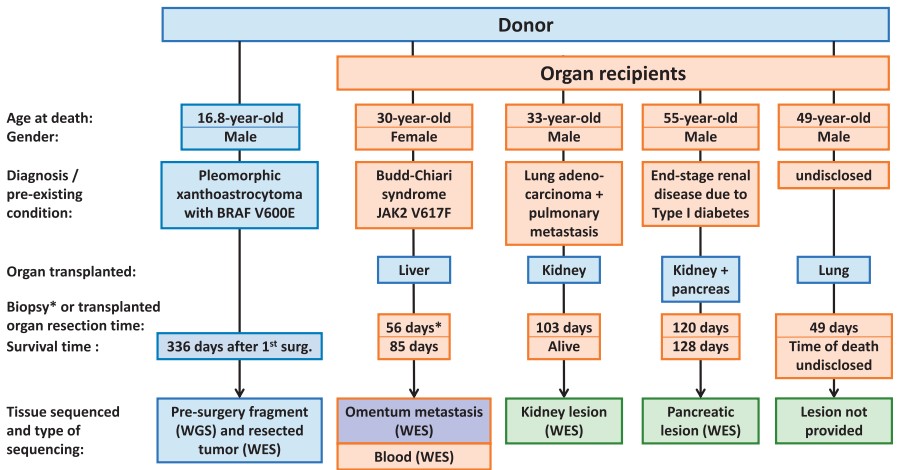

**Figure 1. Case study.**
Clinical data variables for the donor and organ recipients, sample availability, and type of sequencing performed.

**Table 1.   Clinical information.**

|  | Age (yr) | Gender | Event timeline (d) | Survival (d) |
|---|---|---|---|---|
| Donor | 15.5 | male |  |  |
| Craniotomy with evacuation of left temporal hematoma |  |  | −132 |  |
| Second hemorrhage |  |  | −59 |  |
| Hematoma evacuation, tumor fragment |  |  | −10 |  |
| Craniotomy, brain tumor subtotal resection (debulking) |  |  | 0 |  |
| Craniotomy, brain tumor 2nd resection (debulking) |  |  | 56 |  |
| Death | 16.8 |  | 336 |  |
| Liver recipient | 30 | female |  | 85 |
| Liver biopsy |  |  | 56 |  |
| Kidney recipient | 33 | male |  | — |
| Nephrectomy |  |  | 103 |  |
| Kidney pancreas recipient | 55 | male |  | 128 |
| Nephrectomy and a pancreatectomy with resection of the adjacent duodenum, an omentectomy, and biopsies of peritoneal and mesenteric nodules |  |  | 120 |  |
| Lung recipient | 49 | male |  | not disclosed |
| Right upper lobe lesion positive for *BRAF* mutation |  |  | 49 |  |

mellitus. 4 mo after transplantation, a graft nephrectomy and a graft pancreatectomy with resection of the adjacent duodenum, an omentectomy, and biopsies of peritoneal and mesenteric nodules were performed and revealed extensive peritoneal carcinomatosis. Pathology identified a high-grade malignant neoplasm with epithelioid, rhabdoid, and spindle-cell features populating all examined tissues. The patient died 128 d after transplant.

### Lung recipient
The lung recipient was a 49-yr-old male who developed a *BRAF* V600E–positive bronchiogenic carcinoma after transplantation and died. No further information could be obtained.

The samples available for the study were from the donor hematoma evacuation surgery (tumor fragment [TF]) and the first resection surgery (resected tumor [RT]), a blood (LR-B) and an omentum

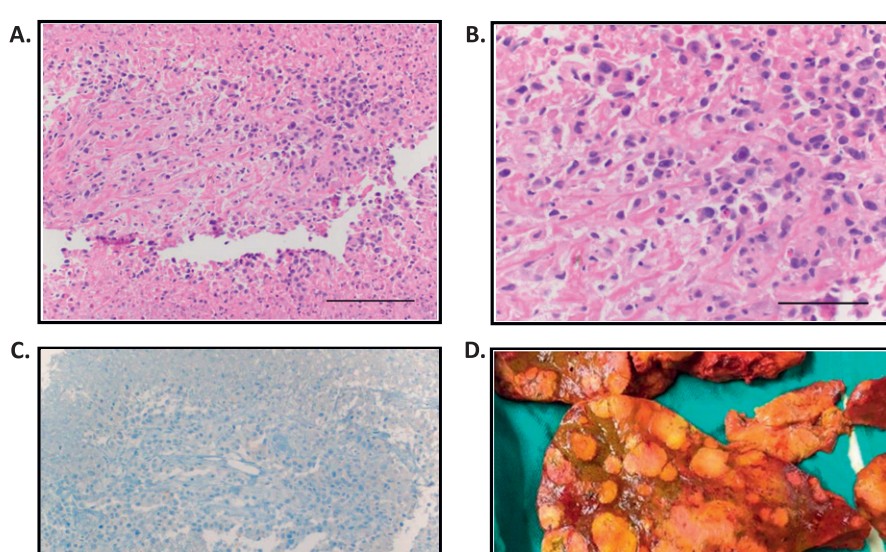

**Figure 2.   Liver recipient.**
**(A, B, C)** Needle biopsy of liver mass: (A) Small focus of viable tumor surrounded by necrosis. Hemotoxylin and eosin stain, magnification scale: 180 $\mu$m. **(B)** Viable tumor shows epithelioid morphology with pleomorphism. The tumor cells have high nuclear to cytoplasmic ratio with small to moderate amount of cytoplasm. The nuclei show large prominent nucleoli. Hemotoxylin and eosin stain, magnification scale: 90 $\mu$m. **(C)** Glial fibrillary acidic protein negative immunostaining (appropriate positive and negative controls were evaluated). Magnification scale: 180 $\mu$m. **(D)** Transversal cut of the liver showing multiple hepatic lesions.

metastasis (LR-Om) samples from the liver recipient, a kidney tumor sample from the KR, and a pancreas sample from the kidney and pancreas recipient.

## Histology, morphology, and immunohistochemistry staining analyses

The cellular morphology of the recipient tumors shared similar features to those of the donor tumor (Fig 2A and B and Table S1). However, glial and neural markers (glial fibrillary acidic protein, S100, or synaptophysin) positively expressed by the donor cells were undetected in the recipient lesions, indicating a possible non glial origin of these tumors (Fig 2C and Table S1). For this reason, several markers were independently tested for each recipient lesion by their respective point of care team to determine their cellular origin. Staining was negative for markers of liver, renal, pancreatic, and lung carcinoma as well as other tumor types including leukemia and lymphoma. A striking difference in the Ki-67 labeling index was also noticeable between the donor tumor and the recipient liver (10% average versus 70%), indicating the high proliferative activity of the latter.

## Tumor lineage

The genetic profiles of the tumors were used to track the lineage from the organ donor, to the participants. The genetic profile of a tumor includes both the DNA of the germline of the affected individual and the somatic mutations acquired over the development and expansion of the cancer. The germline sequence includes both common polymorphisms found in the general population and some rare and/or unique variation that may be specific to this individual. To specifically identify the somatic profile of a tumor, these germline variants, identified by sequencing the individual normal tissue (such as blood), must be subtracted from the tumor sequence. Fig 3A illustrates such an idealized study design for the analysis of tumors in transplant cases in which the donor and the recipient germlines are both known. In some cases, not all tissues will be readily available and tumor samples mixed with non-malignant cells from donor and/or recipient tissue complicate the analyses potentially leading to false assumption. In such situations, extensive DNA sequencing and reference to public databases of known DNA variation can be used to deduce somatic tumor profiles.

In this study, tumor tissue samples were available from the transplant donor, two of four organ recipients, and a metastatic

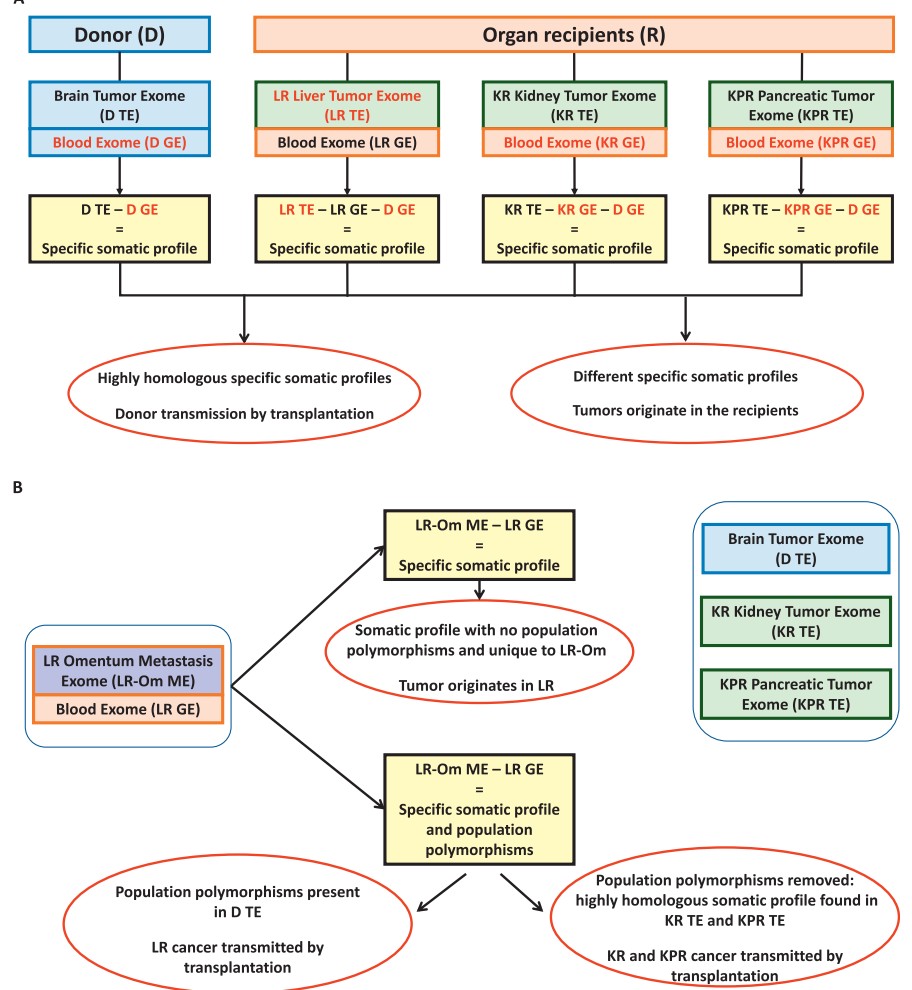

**Figure 3. Possible molecular driven approaches to determine transmission by transplantation.**
**(A)** A straightforward approach consists of establishing the somatic profile of each tumor and determining the degree of homology among these profiles. In such case, the germline exome of the donor and of each recipient is subtracted from each recipient respective tumor exome. **(B)** An approach to consider in the case of limited organ availability consists of using a metastasis, a distinct and distant entity from the transplanted organ, and the blood of such recipient to establish a somatic profile. The recipient germline is subtracted from the metastasis exome. If the metastasis somatic profile obtained is clean and unique when compared with the other recipient exomes, this is not a case of transmission by transplantation. If the profile still contains normal population polymorphisms as listed in the 1000 Genome Project and HapMap cohorts, dbSNP, ExAC, and gnomAD databases in extensive amount that could also be found in the donor tumor exome, this is a case of transmission by transplantation. A "cleaner" somatic profile can be established by filtering out these common polymorphisms and the cancer driver genes identified. If also found in the other lesions, this is a general case of transmission by transplantation.

Table 2. Classification of variants obtained by whole-exome sequencing and amount of potential somatic variants after removal of population common variants.

| Variant classification | Liver Recipient | | | | | Donor | | | Kidney Recipient | | | Kidney Pancreas Recipient | | |
|---|---|---|---|---|---|---|---|---|---|---|---|---|---|---|
| | Omentum metastasis | Blood | Blood germline removed | Common SNPs | Potential somatic variants | Lesion | Common SNPs | Potential somatic variants | Lesion | Common SNPs | Potential somatic variants | Lesion | Common SNPs | Potential somatic variants |
| UTR, unspecified exonic | 9,741 | 8,014 | 1,727 | 1,598 | 129 | 8,674 | 8,497 | 177 | 10,816 | 10,567 | 249 | 11,511 | 11,212 | 299 |
| Indel | 404 | 302 | 102 | 92 | 10 | 333 | 304 | 29 | 414 | 367 | 47 | 446 | 388 | 58 |
| Nonsynonymous (missense) | 14,173 | 10,933 | 3,240 | 3,068 | 172 | 9,860 | 9,699 | 161 | 13,797 | 13,470 | 327 | 14,451 | 14,120 | 331 |
| Nonsense | 115 | 84 | 31 | 25 | 6 | 66 | 63 | 3 | 103 | 91 | 12 | 98 | 89 | 9 |
| Synonymous (silent) | 15,525 | 11,866 | 3,659 | 3,544 | 115 | 11,269 | 11,178 | 91 | 15,658 | 15,457 | 201 | 16,497 | 16,296 | 201 |
| Splice Site | 130 | 97 | 33 | 29 | 4 | 112 | 107 | 5 | 144 | 135 | 9 | 134 | 131 | 3 |
| Non-exonic, ncRNA | 118,464 | 100,074 | 18,390 | 16,924 | 1,466 | 121,712 | 118,933 | 2,779 | 145,911 | 142,632 | 3,279 | 153,175 | 149,074 | 4,101 |
| Total variant count | 158,552 | 131,370 | 27,182 | 25,280 | 1,902 | 152,026 | 148,781 | 3,245 | 186,843 | 182,719 | 4,124 | 196,312 | 191,310 | 5,002 |

lesion from the omentum of the liver recipient. In addition, one blood sample was available from the liver transplant recipient (Fig 1). Using the multi-step approach illustrated in Fig 3B, we sought to establish that the tumor lineage began in the transplant donor and led to the lesions observed in the organ recipients.

First, the omentum metastasis and blood from the liver recipient were analyzed. The liver recipient's germline exome sequence, which was obtained from the recipient's blood sample and contained 131,370 variants, was subtracted from the 158,552 variants in the sequence of the omentum metastasis sample (Table 2). These 27,182 variants that were within the omentum metastasis, but not the organ recipient's germline, should represent a putative somatic mutation profile for the omentum metastasis if the primary tumor originated from the recipient (Fig 3B upper designated section). Alternatively, if the primary tumor originated from the donor, this 27,182 variant set should contain the common and specific donor germline variants as well as somatic variants associated with the development of the donor original PXA tumor (Fig 3B lower designated section). Even without the availability of the donor germline exome, it was possible to determine if this variant set contained at least common population variants.

Population frequency data were then used to distinguish which of the 27,182 variants could be properly ascribed as somatic mutations arising within the tumor genome, or alternatively be germline variants that may have been from the donor's germline. Among the 27,182 variants, 25,280 were already known to be common variants, found in other population studies. The remaining 1,902 variants in the omentum metastasis were therefore the most likely potential somatic variants in this tumor including some donor rare specific germline variants.

The entire set of 27,182 variants in the putative somatic mutation profile for the omentum metastasis was next compared with data from exome sequencing of the donor tumor. Strikingly, the majority of the variants (23,246; 85.5%) were present in the donor tumor, strongly suggesting a common origin of the tumor samples. Among the 25,280 variants classified as "common variants," 22,719 (90%) were seen in the donor tumor data. The majority of the "missing" 10% of common variants (90% of 2,316) were located outside the targeted regions in the exome sequencing and are likely to have been missed by the exome capture reagent in the donor tumor.

The similarity of the donor tumor sequence to that of the omentum metastasis in the recipient was underscored by one other metric. Among the 27,182 variants that made up the somatic profile of the liver recipient's tumor, 20 were located on chromosome Y. The recipient was female, the donor male and these therefore likely reflected donor germline sequences. In aggregate, these data demonstrate that the omentum metastasis was derived from the donor tumor.

The examination of the data from the two other organ recipient's tumor samples provided an additional challenge, when compared to the liver recipient–organ donor comparison, as no blood germline samples were available from either the kidney or the kidney/pancreas recipients. The established relationship between the donor tumor and the liver recipient tumor was therefore leveraged to further clarify the somatic profile of the

omentum metastasis, to enable a more direct comparison with the kidney and kidney/pancreas recipient's tumors.

The KR and KPR recipient tumor exome sequence data were first filtered by removing common population polymorphisms, leaving 4,124, and 5,002 possible somatic variants, respectively (Table 2). These somatic profiles are enriched for somatic events—although most likely still contain some rare germline polymorphisms from each of these individuals as well as the donor. The same process was applied to the donor exome leaving 3,245 variants.

Next, the coding variants of the enriched somatic profiles from all tumors were compared with the coding variants of the somatic profile of the liver recipient's omentum metastasis. We observed 137 coding region variants with an ExAC frequency ≤0.000094 (probably somatic) and 68 variants with an ExAC frequency ≥0.0001 and ≤0.0009 and 33 variants with an allelic fraction (AF) of 0.49–0.51 in the donor tumor (probably rare or specific donor germline variants), all shared among all tumors (Tables S2 and S3), indicating that the KR and KPR lesions were also a case of cancer transmission by transplantation.

## Somatic profile

The 137 somatic coding set shared by all tumors consisted of 82 nonsynonymous, 4 indels, 5 nonsense, 3 splice site, and 43 synonymous coding mutations (Tables 3 and S3). Thirty mutations were predicted to have a deleterious or possible damaging impact on the protein function by at least 8 of the 12 prediction algorithms normally used in sequencing analysis (Table S3). These included the following genes known for their association with cancer: *BRAF* (V600E) and *PIK3CA* (E545K), both harboring hotspot mutations and three other possible driver genes, *SDHC* (H127R), *DDR2* (R668C), and *FANCD2* (C1130Y). The neural cell adhesion molecule (*NCAM1*) gene was affected by a frameshift deletion at the junction of the UTR and coding sequence that resulted in the loss of the transcription start probably affecting the protein integrity. Whole-genome sequencing (WGS) of the donor TF also detected a deletion of over 214,499 bases (chromosome 9: 21,948,801–22,163,300) in the region covering *CDKN2A* and *CDKN2B* (Table 4), Using MLPA, we confirmed the presence of such deletion in the donor RT and recipient lesions (Fig 4). Three major cancer associated pathways were affected in this somatic profile: the activation of the MAPK and phosphoinositide 3-kinase (PI3K) signaling pathways, *BRAF* and *PIK3CA* mutations, respectively, and the inactivation of the tp53 tumor suppressor pathway with the *CDKN2A* and *CDKN2B* deletion.

Other coding events were unique to each lesion or shared among two or three lesions (Tables 3 and S4). Most of those variants were passenger mutations but could also be specific and rare polymorphisms linked to the individual because their exact origin could not be established clearly in the absence of the donor, KR, and KPR germline exome. Only three mutations were in genes linked to cancer development: *NF2* I174fs and *MLL3* both detected in the LR-Om and characterized as pathogenic, and *WT1* in the KR lesion.

*BRAF* is a predominant cancer driver gene, and it can be assumed that its mutation occurred earlier in oncogenesis. So, we used its AF (variant coverage over total coverage) to assess the purity of the tumors (AF fold 2 alleles fold 100), a method often used in pancreatic ductal adenocarcinoma with *KRAS* (Biankin et al, 2012). LR-Om purity was estimated at 97%, KR and KPR lesions at 78%, and the donor TF and RT at 50% and 47%, respectively (Table 4).

*PIK3CA* AF showed a marked increase in the donor's lesions over the 10 d separating the collection of the TF at the time of the hematoma evacuation and the first tumor resection even if both had equivalent tumor purity. Below the level of detection by sequencing, *PIK3CA* variant allelic ratio was estimated to be as low as 1% in the TF by locked nucleic acid (LNA)-PCR sequencing assay (Fig 5), then at 13–14% in the donor RT, equivalent to what had been obtained by Whole exome sequencing (WES). By contrast, it was almost equal to *BRAF* AF in the LR-Om (45% versus 49%) and equal in the KR lesion (both 39%), correlating with the tumor purity of both lesions. This suggests that clonal entities were present in the donor tumor from the time the hematoma was removed. Such clones might have aggressively expanded up to the first surgery and become an active part of the tumor regrowth. However, it cannot be denied that spatial heterogeneity normally present in every tumor could have resulted in sample bias and cause the discrepancy between the two donor tumor samplings even if they had both the same percentage purity. Nevertheless, only the tumor cells containing the *PIK3CA* mutation spread beyond the brain barrier, relocated in different organs, and then aggressively expanded in the immunosuppressed recipients.

**Table 3.** Classification of coding variants shared or unique to a tumor.

| Variant classification | Shared among | | | | | Unique to | | | |
| --- | --- | --- | --- | --- | --- | --- | --- | --- | --- |
| | Four tumors (donor rare germline variants) | Four tumors (somatic variants) | Donor, kidney recipient, KPR lesions | Liver recipient (LR)-Om & two other lesions | Two lesions | Donor lesion | LR-Om metastasis | Kidney recipient lesion | KPR lesion |
| Nonsynonymous (missense) | 69 | 82 | 16 | 6 | 0 | 3 | 46 | 132 | 177 |
| Synonymous (silent) | 29 | 43 | 9 | 2 | 0 | 7 | 42 | 35 | 114 |
| Indel | 2 | 4 | 13 | 0 | 3 | 5 | 3 | 19 | 34 |
| Nonsense | | 5 | 1 | 0 | 0 | 0 | 2 | 11 | 6 |
| Splice site | 1 | 3 | 2 | 0 | 0 | 3 | 1 | 5 | 2 |
| Total | 101 | 137 | 41 | 8 | 3 | 18 | 94 | 202 | 333 |

**Table 4. Variant allelic fraction and estimated tumor cellularity.**

| Gene | Variant classification | Amino acid change | Chr change | Donor TF | Donor RT | Liver recipient (LR)Om | Kidney recipient lesion | KPR lesion | Prediction impact |
|---|---|---|---|---|---|---|---|---|---|
| Estimated Tumor Purity based on *BRAF* V600E allelic fraction | | | | 50% | 47% | 97% | 78% | 78% | |
| *BRAF* | Missense | p.V600E | c.T1799A | 0.25 | 0.23 | 0.49 | 0.39 | 0.39 | High |
| *PIK3CA* | Missense | p.E545K | c.G1633A | bd[a] | 0.14 | 0.45 | 0.39 | 0.18 | High |
| *DDR2* | Missense | p.R668C | c.C2002T | 0.56 | 0.40 | 0.41 | 0.33 | 0.18 | High |
| *SDHC* | Missense | p.H127R | c.A380G | 0.54 | 0.45 | 0.35 | 0.30 | 0.20 | High |
| *FANCD2* | Missense | p.C1130Y | c.G3389A | 0.55 | 0.42 | 0.19 | 0.41 | 0.13 | possibly damaging |
| *NCAM1* | Frameshift Del | p.M1_3del | c.-23_8del | bd[a] | 0.16 | 0.20 | 0.20 | 0.18 | uncertain |
| *NF2* | Frameshift Del | p.I174fs | c.521delT | — | — | 0.20 | — | — | High |
| KMT2C, MLL3 | Missense | p.P3998S | c.C11992T | — | — | 0.07 | — | — | High |
| *WT1* | Stopain | p.R430X | c.C1288T | — | — | — | 0.16 | — | High |
| | | Start position | End position | Methods of detection | | | | | |
| *CDKN2A/B* | Deletion | chr9:21948801 | chr9:22163300 | Whole-genome sequencing | MLPA | MLPA | MLPA | MLPA | |

[a]Below detection.

# Discussion

Here we analyzed tumor DNA from samples derived from a donor and three of the four organ recipients who developed malignancy after transplant. NGS was used for the first time, to track the lineage relationships between the different samples. The analysis of common and rare polymorphisms and cancer associated mutations indicated that cancerous cells were transmitted through transplant. Such cells were the source of the liver recipient's aggressive cancer, and the high mutation profile similarity of the two other recipients' lesions confirms their similar clonal origin.

Anaplastic PXA is a rare tumor and complete specific mutation profiles have not been established. Based on targeted panel sequencing, the most frequent somatic mutations detected in PXA are *BRAF*, *FANCA/D2/I/M*, *PRKDC*, *NF1*, *NOTCH2/3/4*, and *CDKN2A* (Park et al, 2017; Zou et al, 2019). In this study, the tumors were characterized by some of these genetic alterations: *BRAF*, *FANCD2*, and *CDKN2A* found in all lesions. Others observed mutations were not previously associated with anaplastic PXA or PXA: *PIK3CA*, *SDHC*, *DDR2*, and *NCAM*. *PIK3CA* hotspot mutations are usually found in 6–15% of glioblastoma cases where they are linked to increased invasiveness and/or CNS dissemination, early recurrence, and poor prognosis (Tanaka et al, 2019); *SDHC* mutations are found in paragangliomas 3 (PGL3), a neural crest tumor that can develop at various body sites (Niemann & Müller, 2000); and mutations in *DDR2*, a member of the collagen receptor family and a receptor tyrosine kinase, have been identified in a large number of cancers and might play a role in invasiveness (Valiathan et al, 2012; Henriet et al, 2018). The *NF2*, *MLL3*, and *WT1* mutations, genes considered as tumor suppressor genes, were uniquely detected in the LR-Om and KR lesion and might have been acquired post transplantation. Mutations in *NF2* occur in schwannomas and meningiomas, as well as other types of cancer including GBM, hepatic, mesothelioma, breast, colorectal, skin, clear cell renal cell carcinoma, and prostate cancer (Petrilli & Fernández-Valle, 2016). *WT1* is a transcription factor, mutated it has been associated with the development of Wilms' Tumor. Mutations in *MLL3* have been often found in leukemia.

The early regrowth and the observed rapid demise of the donor, which is generally inconsistent with PXA histology usually considered curable, the fast and deadly expansions that were observed in the immunosuppressed hosts, the high homology between the mutation profiles of lesions growing in different organs, the role such mutations might play in other cancers, taken together these observations are consistent with a mutation profile indicative of an aggressive phenotype of clonal entities present in the donor original tumor. Thus, the molecular study of the cohort revealed a logical pathway for the origins and dissemination of this transplanted malignancy.

Precision medicine and the implementation of molecular approaches as a key to decide treatment strategy and predict prognosis is now a reality. But it is still often based on the limited detection of a single mutation such as in this case was *BRAF* V600E. In contrast, NGS technologies offer a global, comprehensive perspective of the somatic mutation profile. As we learn more about the correlation between comprehensive mutation profile and invasion causality, we should seriously consider adding genetic somatic profile

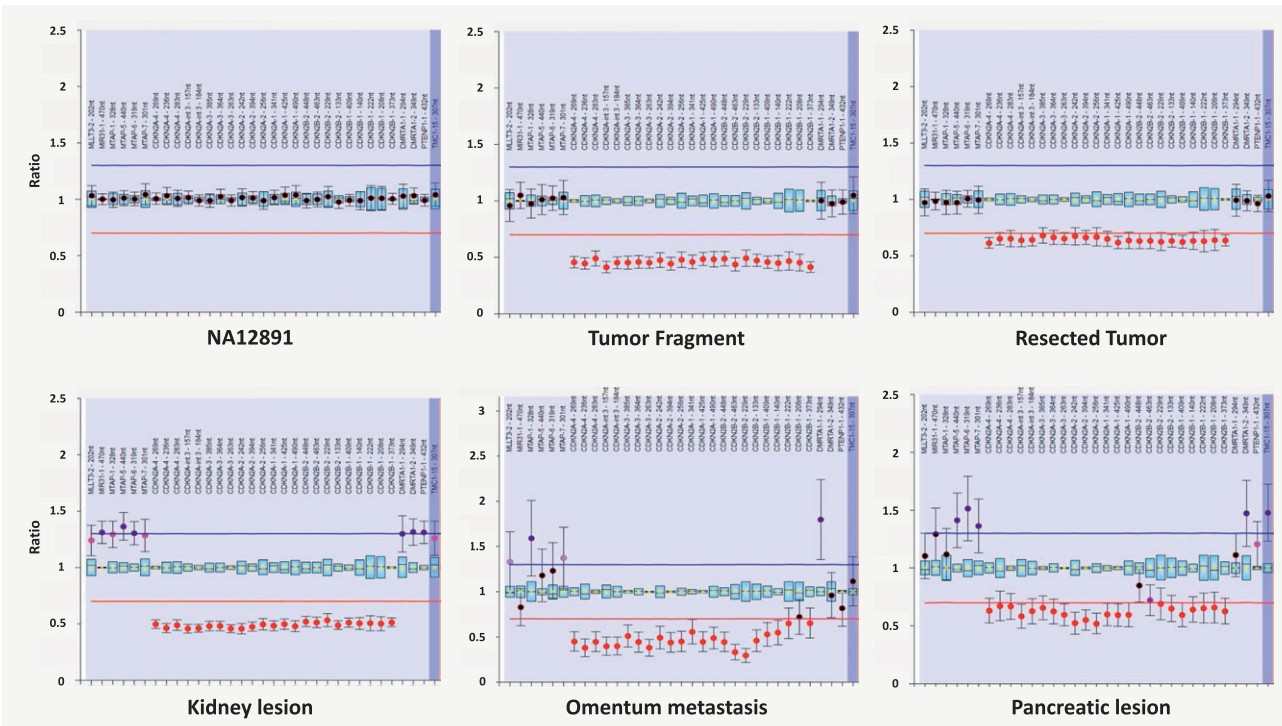

**Figure 4. Ratio chart of the MLPA analysis of chromosomal 9 region covering the genes *CDKN2A* and *B*.**
The dots display the probe ratios and the error bars the 95% confidence ranges. Locations of the region are displayed above the analysis on the x-axis and ratio results on the Y-axis. The red and blue lines at ratio 0.7 and 1.3 indicate the arbitrary borders for loss and gain, respectively. NA12891, one of the three cell lines from the HapMap project used as reference samples, harbors a wild-type allele (ratio of 1) whereas the deletion of the entire region covering the genes *CDKN2A* and *B* is detected in all donor and recipient samples (ratio < 0.7). A higher variation and confidence ranges is apparent in the formalin-preserved omentum metastasis of the liver recipient and in the pancreatic formalin-fixed paraffin-embedded lesion of the kidney and pancreas recipient in comparison to the other fresh frozen solid tissues (donor and kidney recipient samples) because of the chemical treatment of the tissues.

to the list of consideration of transplant safety. This is especially feasible when tumors are removed months before the donor deadly outcome, and original and regrowth tumors can be analyzed for mutation profile evolution and aggressiveness.

Comparatively, methods such as the detection of circulating tumor cells or somatic DNA to assess potential metastasis face a high challenge in the case of brain tumor such as but not limited to the presence of the brain barrier limiting their amount in circulation and the lack of reliable tumor markers. In this case, for example, glial and neural markers (glial fibrillary acidic protein, S100, or synaptophysin) were positively expressed by the donor tumor cells but were undetected in the recipient lesions.

## NA12891 HapMap cell line DNA and DNA spiked with mutated DNA

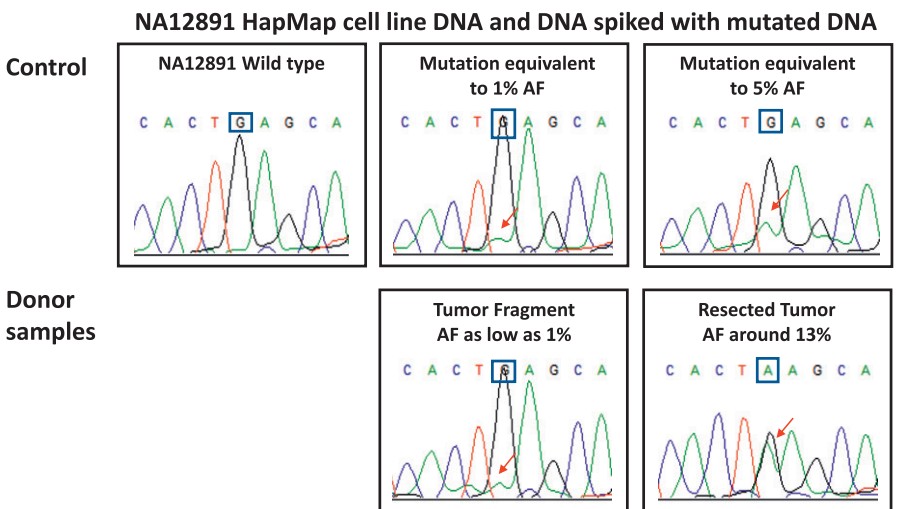

**Figure 5. *PIK3CA* G1633A (E545K) locked nucleic acid (LNA)–PCR sequencing.**
The HapMap project NA12891 cell line harboring a wild-type allele (*PIK3CA* c.G1633G, □) was used as a normal control. The NA12891 DNA was also spiked with the mutated DNA of the kidney lesion at an equivalence of 1% and 5% variant allelic fraction to establish baseline controls. Based on these control chromatograms, the *PIK3CA* mutation was present in the donor resected tumor as well as in the tumor fragment but at an allelic fraction as low as 1% in the tumor fragment and around 13% in the resected tumor.

There is a significant mismatch between the increasing demand of tissues for transplant and donor organ availability. Although deceased organ donors have steadily increased since 2011, only ~3 in 1,000 people die in a manner that presently allows for organ donation (https://optn.transplant.hrsa.gov/data/view-data-reports/national-data/, https://www.organdonor.gov). There is a critical need for additional strategies to narrow this substantial gap. By ensuring the safety of organs to be transplanted, the NGS-based technologies can discard some donors but conversely increase the pool of individuals that were rejected uniquely on the basis of cancerous condition.

Transplant from deceased donors requires careful screening of donor and organ integrity via methods that do not delay surgery. NGS methods can provide precise molecular pathology profiles for tumors observed in donors from sites distal to the organs to be transplanted and can therefore reduce the likelihood of cancer transmission and possibly increase the donor pool. As the development of more rapid turn-around times for NGS assays proceeds, molecular technologies can be applied to insure safer transplantation of well-preserved organs.

# Materials and Methods

### Histology, morphology, and immunohistochemistry staining

The donor and recipient hemotoxylin and eosin (H&E)–stained slides from the formalin-fixed paraffin-embedded (FFPE) lesions were examined by the pathologists from the original sites of collection. Immunohistochemistry analysis for the detection of subtype markers expression was performed at each site.

### Samples

We acquired: solid frozen tumor samples from the donor hematoma evacuation surgery and the first resection surgery; and the KR; a blood sample collected in PAXgene Blood Tube and an omentum metastasis sample preserved in a formalin solution from the liver recipient; and a FFPE pancreas sample from the kidney and pancreas recipient. Sample from the lung recipient was not available. The study was performed under a Baylor College of Medicine Institutional Research Board approval protocol H21497.

### Nucleic acid isolation

DNA was isolated in accordance to the tissue type as per manufacturer protocol: LR blood draw in PAXgene Blood Tube: PAXgene blood DNA kit (Cat no. 761133; PreAnalytiX/QIAGEN); donor and KR solid tissue lesions: Gentra Puregene (Cat no. 158667; QIAGEN); LR-Om in formalin and FFPE sections from KRP pancreatic lesion: QiaAmp DNA FFPE tissue kit (Cat no. 56404; QIAGEN). DNA was quantified using the Quant-iT PicoGreen dsDNA Assay kit (Cat no. P11496; Thermo Fisher Scientific).

### Whole-exome sequencing

WES was performed on the NovaSeq 6000 platform on the donor RT, LR blood and metastasis, and KR and KPR lesions.

#### *WES library preparation*

DNA from the different sources was constructed into Illumina paired-end pre-capture libraries according to the manufacturer's protocol (Multiplexing SamplePrep Guide 1005361D; Illumina) with modifications as described in the BCM-HGSC Illumina Barcoded Paired-End Capture Library Preparation protocol.

Briefly, 200 ng of DNA in 50 μl volume was sheared into fragments of average size of 300 base pairs in a Covaris plate with E220 system (Covaris, Inc.) followed by end-repair (End Repair module, E6050; New England BioLabs [NEB]), A-tailing (NEBNext dA-Tailing Module, E6053; NEB), and ligation of the Illumina multiplexing PE adaptors (ExpressLink Ligase, Cat no. A13726101; Invitrogen).

Pre-capture Ligation Mediated-PCR (LM-PCR) was performed for six cycles using the Library Amplification Readymix containing KAPA HiFi DNA Polymerase (Cat no. KK2612; Kapa Biosystems, Inc.). Universal primer LM-PCR Primer 1.0 and LM-PCR Primer 3.1 were used to amplify the ligated products. Reaction products were purified using 1.8× Agencourt AMPure XP beads (Cat. no. A63882; Beckman) after Covaris shearing, end-repair, and A-Tailing. After adapter ligation and PCR amplification, libraries were purified twice using 1.2× Agencourt AMPure XP beads. After the final XP bead purification, quantification and size distribution of the pre-capture LM-PCR product was determined on Agilent Bioanalyzer 2100 using the DNA7500 kit (5067-1506; Agilent) (Table 5).

#### *Exome capture*

The FFPE and Non-FFPE samples were pooled separately using 500 ng of library for each sample. These two pools of libraries were then hybridized in solution to the HGSC VCRome 2.1 design (Bainbridge et al, 2011) (42 Mb [mega base]; NimbleGen) according to the manufacturer's protocol *NimbleGen SeqCap EZ Exome Library SR User's Guide* (*Version 2.2*) with minor revisions. For ~3,500 clinically relevant genes that had low coverage (<20× coverage at ~2.72 Mb sequencing data) probes were supplemented with PKv1 and PKv2

**Table 5.  Library yield details for whole-exome sequencing.**

| Sample | Library average size (bp) | Library yield (ng) | Pool |
|---|---|---|---|
| Liver recipient Omentum metastasis | 405 | 1,124.3 | Formalin-fixed paraffin-embedded Pool |
| KPR Pancreatic lesion | 436 | 1,737.5 | |
| Kidney recipient Kidney lesion | 468 | 2,384 | |
| Donor Resected Tumor | 470 | 2,297 | Non formalin-fixed paraffin-embedded Pool |
| Liver recipient Blood | 461 | 2,633.3 | |

**Table 6.  Whole-exome sequencing sequencing metrics.**

| Sample | Total MB | Unique Aligned MB | Percent Unique | Percent Duplicate Reads | Median Insert Size | Average Coverage | % base covered 1× | % base covered 20× |
|---|---|---|---|---|---|---|---|---|
| Donor resected tumor | 21,565 | 18,497 | 85.9 | 19.99 | 304 | 199 | 99.83 | 99.21 |
| Liver recipient omentum metastasis | 11,757 | 17,155 | 82.8 | 22.81 | 208 | 113.67 | 99.64 | 98.29 |
| Liver recipient blood | 19,832 | 9,726 | 86.59 | 18.59 | 298 | 187.64 | 99.77 | 99.02 |
| Kidney recipient kidney lesion | 21,752 | 18,171 | 83.67 | 23.07 | 303 | 191.15 | 99.83 | 99.22 |
| KPR pancreatic lesion | 30,426 | 23,328 | 76.76 | 34.89 | 268 | 240.94 | 99.82 | 99.33 |

reagent spiked into the VCRome 2.1. Human COT1 DNA and xGen Universal Blocking oligonucleotides (Integrated DNA Technologies) were added into the hybridization to block repetitive genomic sequences and the adaptor sequences and hybridization was carried out at 42°C for 72 h. Post-capture LM-PCR amplification was performed using the Library Amplification Readymix containing KAPA HiFi DNA Polymerase (Cat no. KK2612; Kapa Biosystems, Inc.) with 12 cycles of amplification. After the final AMPure XP bead purification, quantity and size of the capture library was analyzed using the Agilent Bioanalyzer 2100 DNA Chip 7500.

### WES
Sequencing was performed on the NovaSeq 6000 instrument using the S4 reagent kit (300 cycles) to generate 2 × 150 bp paired-end reads. Post-capture library pools were sequenced on NovaSeq S4 flow cell to generate between 9.7 and 23.3 Gb unique sequence data per sample (Table 6).

### WGS

WGS was performed on the HiSeq X platform on the donor TF to identify major deletions and insertions and see the status of ascertained driver genes and their AF in this lesion by comparison to the others.

### WGS library preparation and sequencing
WGS libraries were prepared as reported earlier (Kessler et al, 2020; Raffield et al, 2020) and sequenced in two lanes on HiSeq X to generate 77.5× sequence coverage (Table 7).

### Data analysis

### Primary data analysis
Initial sequence analysis was performed using the HGSC Mercury analysis pipeline (https://www.hgsc.bcm.edu/software/mercury) as follows. The .bcl files produced by the primary analysis software were

transferred into the HGSC analysis infrastructure by the HiSeq Real-time Analysis module. Mercury ran the vendor's primary analysis software (CASAVA) which demultiplexed pooled samples and generated sequence reads and base-call confidence values (qualities). The reads were then mapped to the GRCh37 Human reference genome (http://www.ncbi.nlm.nih.gov/projects/genome/assembly/grc/human/) using the Burrows-Wheeler aligner (http://bio-bwa.sourceforge.net/) (Li & Durbin, 2009) to produce a binary alignment/map (BAM) file (Li et al, 2009). The last step involved quality recalibration (using GATK [DePristo et al, 2011] https://www.broadinstitute.org/gatk/) and, where necessary, the merging of separate sequence-event BAMs into a single sample-level BAM. BAM sorting, duplicate-read marking, and realignment to improve in/del discovery all occurred at this step.

### Cancer data analysis
Primary BAM files were separately run through Atlas-SNP (Shen et al, 2010) and PinDel (Ye et al, 2009). Variant annotation was performed using Annovar (Wang et al, 2010), COSMIC (Forbes et al, 2011), and dbSNP (Sherry et al, 2001). Low-quality variants and normal population polymorphism calls were removed by filtering against the 1000 Genome Project and HapMap cohorts, dbSNP, ExAC, and gnomAD databases for variant present in the population at an allele frequency greater or equal to 0.001 in more than one database. Prediction of the mutation impact on the protein function was performed using 12 different prediction software (SIFT4G, Polyphen2 HDV, Polyphen2 HVAR, Mutation Taster, DEOGEN2, M-CAP, ClinPred, fathmm-MKL coding, fathmm-XF_coding, BayesDel_addAF, BayesDel_noAF, and LIST-S2).

### Structural variant analysis
Structural variants (SVs) were identified from the WGS BAM file using Parliament2, an ensemble SV caller, that runs a combination of tools to generate SV calls on whole-genome sequencing data, followed by genotyping step using SVTyper and merging step using SURVIVOR tool (https://github.com/dnanexus/parliament2 [English et al, 2015]). SV calls were annotated using an in-house script and UCSC gene models. SVs were filtered to include only deletions that

**Table 7.  Whole-genome sequencing metrics.**

| Sample | Yield bases | Percent Aligned bases | Percent Duplicate Reads | Median Insert Size | Average Coverage | % base covered 1× | % base covered 20× |
|---|---|---|---|---|---|---|---|
| Donor tumor fragment | $2.55 \times 10^{11}$ | 97.81 | 9.14 | 422 | 77.56 | 100 | 99.74 |

had genotyped calls, were identified by more than one underlying algorithm, and overlapped coding exons. We used an internal SV dataset from 41 samples to filter out variants common in general population. Finally, the resulting set of rare deletions was furthered filtered to include variants overlapping 596 cancer genes.

### Locked nucleic acid PCR sequencing assay

*PIK3CA* G1633A (E545K) low AF mutation was detected in the original donor lesion by LNA PCR sequencing assay (Ang et al, 2013). Briefly, 100 ng of DNA was used to amplify the mutated *PIK3CA* region using HotStarTaq DNA Polymerase (QIAGEN), PCR primers, and thermal cycling parameters as specified by Ang et al (2013). A LNA probe binding to the reference sequence around *PIK3CA* codon 545 was added to the Sanger reaction to enrich the yield of mutant alleles. Wild-type DNA (Coriell 1000 Genome Project sample NA12891) and KR mutated DNA spiked into the wild-type DNA at concentrations equivalent to an allelic ratio of 5% and 1% were used as positive controls.

### Multiple ligation-dependent probe amplification (MLPA) assay

The MLPA assay was conducted to assess the copy number variation of the *CDKN2A* genomic region according to the manufacturer's protocols (MRC-Holland). In brief, 50 ng of control DNA (HapMAp project NA12878, NA12891, and NA12892 cell lines, used as reference samples), 50 ng of DNA extracted from the donor and KR solid tissue lesions or 100 ng of DNA extracted from the formalin-preserved LR-Om and FFPE KRP pancreatic lesion were denatured at 98°C for 5 min and subsequently hybridized to the MLPA probe mix (P419-CDKN2A/2B-CDK4, MRC-Holland) at 60°C for 16 h. After hybridization, the ligase was added in the reaction tube. The ligation reaction was performed at 54°C for 15 min followed by inactivation of the ligase at 98°C for 5 min. Ligated probe pairs were amplified as follow: 98°C 5 min, 35 cycles of (95°C 30 s, 60°C 30 s, 72°C 60 s), and a final incubation at 72°C 20 min. The PCR amplified fragments were separated by capillary electrophoresis using the ABI 3500 Genetic Analyzer (Applied Biosystems) with GS600 size standard. The MLPA data were analyzed with the Coffalyser.net Software (MRC-Holland).

## Data Availability

The datasets generated during the current study have been deposited in dbGAP.

## Supplementary Information

## Acknowledgements

The authors acknowledge Allison Boyer at Life Share of Oklahoma, the pathology teams of the different institutions who graciously send us tissues, Dr Patricia Castro, Director of the Human Tissue Acquisition and Pathology at Baylor College of Medicine, and the people at the Human Genome Sequencing Center who made the realization of this study possible. This work was supported by a grant from the National Human Genome Research Institute.

## Authors' Contributions

M-C Gingras: conceptualization, data curation, formal analysis, and writing—original draft, review, and editing.
A Sabo: formal analysis.
M Cardenas: formal analysis.
A Rana: resources.
S Dhingra: formal analysis.
Q Meng: formal analysis and supervision of methodology.
J Hu: formal analysis and supervision of methodology.
D Muzny: supervision of methodology.
H Doddapaneni: supervision of methodology.
L Perez: methodology.
V Korchina: methodology.
C Nessner: methodology.
X Liu: methodology.
H Chao: methodology.
J Goss: conceptualization.
RA Gibbs: conceptualization, funding acquisition, and writing—review and editing.

## Conflict of Interest Statement

The authors declare that they have no conflict of interest.

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
