## [Reviewer comments · Life Science Alliance]

Life Science Alliance

Sequencing of a central nervous system tumor demonstrates cancer transmission in an organ transplant

Marie-Claude Gingras, Aniko Sabo, Maria Cardenas, Abbas Rana, Sadhna Dhingra, Qingchang Meng, Jianhong Hu, Donna Muzny, Harshavardhan Doddapaneni, Lesette Perez, Viktoriya Korchina, Caitlin Nessner, Xiuping Liu, Hsu Chao, John Goss, and Richard Gibbs

DOI: <https://doi.org/10.26508/lsa.202000941>

Corresponding author(s): Marie-Claude Gingras, Baylor College of Medicine

Review Timeline:

Submission Date:	2020-10-20
Editorial Decision:	2021-01-28
Revision Received:	2021-05-05
Editorial Decision:	2021-06-01
Revision Received:	2021-06-29
Editorial Decision:	2021-07-06
Revision Received:	2021-07-09
Accepted:	2021-07-12

Transaction Report:

January 28, 2021

Re: Life Science Alliance manuscript #LSA-2020-00941-T

Dr. Marie-Claude Gingras
Baylor College of Medicine
Human Genome Sequencing Center
One Baylor Plaza, MSC226
Houston, TX 77030-3411

Dear Dr. Gingras,

Thank you for submitting your manuscript entitled "How DNA sequencing of tumors from the central nervous system can lead to safer organ transplantation" to Life Science Alliance. The manuscript was assessed by expert reviewers, whose comments are appended to this letter.

We apologize for this delay in getting back to you. It took us longer than expected to fulfill the full panel of reviewers for this study. As you will note from the reviewers' comments below, for the most part, the reviewers were intrigued by these findings. However, they have expressed some concerns over some conclusions not being supported by the findings and requested additional data that should improve the robustness of the manuscript. We would thus request you to submit a revised version of this study back to us that addresses all of the reviewers' points, either by toning down some of the conclusions (as pointed out by Rev 2 and 3), improving data presentation (Rev 3) and with additional data (requested by all reviewers).

Thank you for this interesting contribution to Life Science Alliance. We are looking forward to receiving your revised manuscript.

Sincerely,

Shachi Bhatt, Ph.D.
Executive Editor
Life Science Alliance
<https://www.lsjournal.org/>
Tweet @SciBhatt @LSAJournal

- A letter addressing the reviewers' comments point by point.
- An editable version of the final text (.DOC or .DOCX) is needed for copyediting (no PDFs).
- High-resolution figure, supplementary figure and video files uploaded as individual files: See our detailed guidelines for preparing your production-ready images, <https://www.life-science-alliance.org/authors>
- Summary blurb (enter in submission system): A short text summarizing in a single sentence the study (max. 200 characters including spaces). This text is used in conjunction with the titles of papers, hence should be informative and complementary to the title and running title. It should describe the context and significance of the findings for a general readership; it should be written in the present tense and refer to the work in the third person. Author names should not be mentioned.

B. MANUSCRIPT ORGANIZATION AND FORMATTING:

Reviewer #1 (Comments to the Authors (Required)):

The authors of "How DNA sequencing of tumors from the central nervous system can lead to safer organ transplantation" performed a valuable, interdisciplinary investigation regarding the emerging topic of donor/recipient cancer transmission after organ transplantation. The authors provide a comprehensive asset of data in which authors claim to have enough evidence to support the

observation of cancer transmission in 3 patients recipient of 4 organs from a person affected by a fatal malignancy called anaplastic pleomorphic xanthoastrocytoma (APX).

The study raises important issues in the current medical practice and includes a complex computational analysis of tumor tissues' genome appertaining to 4 individuals. Sequencing experiments are followed by experimental validation of one of the main findings, deletion of CDKN2A/B gene, which is present in the donor tissue, and distal metastasis of 1 patient.

The main concern raising to this reviewer is related to the fact that only one distal metastasis has been sequenced in the entire study while the other genomic profiles are from the donor tissues. Thus, it is not surprising to find the same SNPs assets.

The paper also needs a more detailed description of the tissues' genomic profiling instead of describing only conclusions. This reviewer finds challenging to link the results from WES of KR and PKR to the LR-Om since they are entirely different tissues in which the latter should represent a mix of recipient/donor mixture of cells instead of donor-specific cells.

Overall, the study is of interest for the clinical evidence of tumor growth in distal tissues from the brain tumor of origin once transplanted in 3 different patients. However, the study lacks the focus needed to demonstrate what is described in the abstract.

The main concerns raised by the reviewer are the following:

1. The authors performed genomic profiling by WGS and WES as described in FIG.1. However, the description of data analysis in section "Mutational profile and comparative genomic analyses" is highly deconvoluted and connects different pieces of information without a clear link. Ex: The authors did not describe the WES of each tissue but directly jump to the conclusion of the analysis which should be described in detail. Priority needs to be assigned to the description of every tissue profiling by identifying SNPs and the variant frequency of each tissue profiling and then describing (also visually) the commonalities.
2. It is clear the rationale behind the LR-Om sequencing; however, not enough description is provided to the sequencing of the other patient tissues.
3. The authors need to explain why they did not sequence metastasis of the patient KR and PKR.
4. Pag 8. contained a total of 32,216 variants, of which 24,194 were LR germline variants (found in the exome of the blood cells). How many variants are generally found in a germline analysis? How many of them are SNPs?
5. From tables 2 and 3, 55 rare variants are found. The authors should declare the meaning of "rare".
6. Was BRAF-V600 mutation a germline or somatic mutation?
7. What is the calculation to estimate tumor purity based o BRAF V600E?
8. According to the journal scope, this sentence "The datasets generated during the current study are available from the corresponding author on reasonable request" should be deleted and raw and

downstream data uploaded on GEO or ENA database.

Reviewer #2 (Comments to the Authors (Required)):

1. The authors tried to track the relationship of the donor and the recipient's tumors using next generation sequencing technology. NGS was used to profile whole genome and/or whole exome of the donor and the recipient's tumor and germline. They identified 55 rare variants and 156 coding somatic mutations that were shared among the donors and the recipients' lesions. They also identified a chromosomal deletion spanning CDKN2A/B genes that were shared among the samples. The paper concludes that the donor/recipient cancer transmission was confirmed at a molecular level based on the similarity of the mutational profiles of the donor and recipient samples and the recipient tumor is a clonal expansion of the donor tumor.

2. Transmission through transplantation of an organ with undetected metastasis has been known to occur as shown in various previous publications. So it is essential to show through mutation profiling that there are cancer cells or cancer causal mutation in the donor organs. In this paper, it is not clear whether the transmission was caused by the undetected tumor cells or causal mutations in the transplanted organ. Is it possible to identify/separate donor organ somatic mutations from the recipient cancer tissue and establish the transmission path from the original donor's tumor to the donor's healthy transplant organ and finally to the recipient.

3. The authors did great work in profiling and identifying the somatic mutations of the tumor samples and tracking the origin of the recipient's cancer, but failed to show how the donor's tumor genome sequencing information can be used to enhance organ transplant safety as the paper's title claimed.

Reviewer #3 (Comments to the Authors (Required)):

The authors investigate a case study of cancer transmission through transplantation, focusing on the genetic profiles of 4 tumors: the donor's tumor sample, and 3 (of 4) recipient's tumor samples. The introduction describes the state of organ donation from donors with CNS tumors, and the overall low risk for transplantation of non-CNS organs from this type of donor. The anaplastic pleomorphic xanthoastrocytoma (PXA) CNS tumor of the donor was not expected to metastasize outside of the CNS; nevertheless, aggressive neoplasms were detected in the recipients shortly after transplantation. They note previous studies of transmission of cancer through transplantation, where origin of the cancers is primarily verified by histology and analysis of markers found in the donor's tumor. The authors propose 2 questions to be answered in this study: (1) can the origin of the recipients' tumors be determined by NGS analysis? and (2) does this open a new perspective on using NGS methods to reduce the likelihood of transmission at transplantation? The first question is sufficiently addressed in the paper, but the second question (and the related title of the manuscript) is primarily speculation and related claims should be toned down accordingly. The results describe each of the patients and neoplasms in the study in great detail, including their clinical characteristics and histology/morphology/immunohistochemistry analysis. They then describe a NGS analysis of the donor and recipient tumors, which appears to be the first time this technology has been applied to investigate transplant-associated neoplasms. The NGS analysis of the tumors focuses primarily on comparing the omentum metastasis of the liver recipient patient to the blood

sample of the liver recipient patient. This choice seems driven by the availability of samples (i.e. no germline sample from the donor or other recipients). By comparing this metastatic sample to the germline of that patient, the authors are able to identify variants unique to the cancerous cells in that patient, though they cannot technically distinguish rare variants from the donor's germline from somatic variants unique to the cancer cells. By filtering on population frequency, the authors define a "somatic set", demonstrate that these variants are present in the donor's tumor and the tumors of 2 other recipients, and highlight some of the cancer-associated genes mutated (BRAF, PIK3CA, SDHC, FANCD2, DDR2, NCAM1). They also identify a chromosome 9 deletion in the donor tumor through whole genome sequencing, and verify the presence of this deletion in the other tumor samples using MLPA. The authors briefly mention variants unique to each lesion or shared by a subset of lesions, but do not quantify or include details on these. Further investigation into the PIK3CA variant is shown, due to the discrepancy in its detection between the first and second donor samples. The authors argue that the higher PIK3CA frequency in the resected sample indicates aggressive growth of this clone in the 10 days between when the samples were taken. The authors provide sufficient evidence that NGS-based mutation profiles have the potential to confirm the origin of tumors that arise after transplantation. The authors discuss the mutations found in this tumor sample that are not typically associated with PXA, and claim that their association with other more aggressive neoplasms could provide an explanation for the transmission of cancer in this case.

I have the following concerns and suggestions that should be addressed to improve the manuscript:

1. Figure 1 and the related results section has several issues that need improvement. The legend reads like a results section and does not actually describe what is shown in the figure flowchart. The last 2 rows, "tissue sequenced" and "exome association" are confusing and possibly inaccurate. For example, there is mention in the figure of recipient blood for the kidney and pancreas patients, but it does not appear that was sequenced. It is generally difficult to parse from this figure and the text exactly what tissues were sequenced with what technologies. The text of the "mutational profile and comparative genomic analyses" is confusing overall and should be rewritten. For example, the authors state: "None of the lesions growing inside the transplanted organs could be used since they were infiltrated by the donor normal cells, hence the donor germline, which could result in false assumption", but then go on to "use" these samples in the analysis.
2. The analysis focuses on the "somatic set" of variants identified in the omentum sample which is shared between the donor tumor and the recipient tumors. The authors should quantify in the text, and include details in a supplemental table, of all variants identified that were NOT shared between all samples (i.e. the NF2 and NOTCH4 variants). Without this, the reader cannot assess the similarity of the profiles, since only the agreements are described in detail. The lack of clarity on their precise origin does not preclude the authors from describing what they do know about these variants.
3. The use of ExAC frequency to distinguish between "rare variants" and "coding somatic mutations" is a reasonable step but is not conclusive. True somatic variants could have frequencies in the range indicated for rare variants, and rare variants could be absent from the ExAC database. It would be more accurate to call this set "likely somatic".
4. This study would have been more straightforward if a germline sample from the donor was available. In that case, somatic variants could have been easily identified and evaluated in all other samples. The authors should address this limitation in the study design.
5. To strengthen the analysis and evidence of the similarity of the mutation profiles between tumors, the authors could show a global comparison of allele frequencies (i.e. a scatter plot of the allele frequency columns in the supplemental tables).
6. The conclusion that the higher PIK3CA frequency in the resected sample compared to the fragment sample indicates aggressive growth of this clone in the 10 days between when these

samples were taken does not consider alternative explanations. Spatial heterogeneity of the PIK3CA subclone and sampling bias of the two samples could also explain the discrepancy. The authors use this evidence to support their claim that repeated NGS sampling and evolution of a mutation profile could indicate aggressiveness and potential spread of the tumor, but this is mostly unfounded speculation based on this one data point. The authors should discuss alternative explanations for this finding.

7. Linking several mutations found in this PXA case with an aggressive phenotype based purely on the presence of these gene mutations in other malignancies is not well supported by the literature and the current understanding in the field. Using these mutations as a convenient explanation for the aggressiveness of this case and suggesting that this same prediction could be made in future cases without knowing the outcome is a little misleading. The authors suggest that genetic profiling will be useful as "we learn more about correlation between comprehensive mutation profile and invasion causality", which is likely true, but I find their conclusions specific to this case to be overstated based on current knowledge.

8. Minor figure suggestions: the Figure 3 axis labels and text is small and difficult to read. The legend does not describe what is actually plotted (what do the colors represent, what are the error bars, etc.). Table 2 should indicate which samples this list of events are from.

9. In the authors' discussion of the potential for NGS to assist in transplant decisions in the future, they could consider discussing detection of circulating tumor cells or cell-free tumor DNA as a means of assessing potential metastasis.

10. Sequencing data should be deposited in a repository (dbGAP, EGA) as is standard in the field for exome and whole genome sequencing.

Reviewer #1 (Comments to the Authors (Required)):

The authors of "How DNA sequencing of tumors from the central nervous system can lead to safer organ transplantation" performed a valuable, interdisciplinary investigation regarding the emerging topic of donor/recipient cancer transmission after organ transplantation. The authors provide a comprehensive asset of data in which authors claim to have enough evidence to support the observation of cancer transmission in 3 patients recipient of 4 organs from a person affected by a fatal malignancy called anaplastic pleomorphic xanthoastrocytoma (APX).

The study raises important issues in the current medical practice and includes a complex computational analysis of tumor tissues' genome appertaining to 4 individuals. Sequencing experiments are followed by experimental validation of one of the main findings, deletion of CDKN2A/B gene, which is present in the donor tissue, and distal metastasis of 1 patient.

The main concern raising to this reviewer is related to the fact that only one distal metastasis has been sequenced in the entire study while the other genomic profiles are from the donor tissues. Thus, it is not surprising to find the same SNPs assets.

The paper also needs a more detailed description of the tissues' genomic profiling instead of describing only conclusions. This reviewer finds challenging to link the results from WES of KR and PKR to the LR-Om since they are entirely different tissues in which the latter should represent a mix of recipient/donor mixture of cells instead of donor-specific cells.

Overall, the study is of interest for the clinical evidence of tumor growth in distal tissues from the brain tumor of origin once transplanted in 3 different patients. However, the study lacks the focus needed to demonstrate what is described in the abstract.

The main concerns raised by the reviewer are the following:

1. The authors performed genomic profiling by WGS and WES as described in FIG.1. However, the description of data analysis in section "Mutational profile and comparative genomic analyses" is highly deconvoluted and connects different pieces of information without a clear link. Ex: The authors did not describe the WES of each tissue but directly jump to the conclusion of the analysis which should be described in detail. Priority needs to be assigned to the description of every tissue profiling by identifying SNPs and the variant frequency of each tissue profiling and then describing (also visually) the commonalities.

We acknowledge that the previous submission was convoluted and we have endeavored to be clearer in this new submission. The convolution is largely because of the technical challenges of the study, which aimed to determine if the somatic profile of different tumors in different transplant recipients was commonly shared, and to demonstrate that the tumor origins can all be traced to a single tissue donor. Due to the limited availability of tissues and the practical issues associated with distinguishing somatic mutations from recipient and donor germline variants that could originate from the tumor cellular makeup or from contamination of infiltrating recipient blood cells and donor cells present in the transplanted organ, we applied a series of DNA sequence-based approaches. We generated a somatic

mutation profile of a metastasis distant from the transplanted organ in a recipient from which we could establish the germline exome that was then subtracted from the metastasis exome. We also utilized population frequency data to ascertain whether the somatic profile was 'clean' and could be properly attributed to the tumor genome – or else whether the presumed somatic profile contained many variants that were common population polymorphisms. The latter would indicate that the presumed somatic profile was in fact including DNA polymorphisms that were from the donor which could be confirmed by comparing them to those found in the donor tumor exome.

In the case of the liver recipient omentum metastasis, we obtained a somatic profile that still contained a large number of polymorphisms commonly found in the population, including variants on the Y chromosome whereas the recipient was a female. These polymorphisms were present as well in the donor tumor exome, indicating that the tumor was transmitted by the donor at transplantation.

The population polymorphisms were then subtracted from the metastasis exome to obtain a “purified” somatic profile. The presence of this “purified” somatic profile in the donor tumor exome, reinforced the proof of the tumor origin. Similarly, its presence in the tumor exome of the two other organ recipients proved their origin by deduction.

We understand that our approach and deduction were not clearly explained for the reader to follow. For this reason, the **“Mutational profile and comparative genomic analyses”** section was split in 2 new sections and several explanatory paragraphs were added as well as a new Figure (Fig 3) and modified Figure 1. A new table summarizing the number of variants/polymorphisms found in the exome of each tumor and the blood of the liver recipient, and illustrating the number of population polymorphisms that could still be found in the metastasis exome after subtraction of the blood exome was added. We also analyzed how many common population polymorphisms were in the other lesions and subtracted them. We modified table 2 to include the distribution of the variants among the 4 lesions (common to 4, 3, or 2 lesions, or unique to 1 lesion) and we listed them in a new supplemental table S4.

2. It is clear the rationale behind the LR-Om sequencing; however, not enough description is provided to the sequencing of the other patient tissues.

As mentioned above, we have added several tables and description that we hope will provide more information and a better understanding of the study design and results.

3. The authors need to explain why they did not sequence metastasis of the patient KR and PKR.

The other patients were treated in different institutions in other states, and we could only use the tissues provided to us through LifeShare.

4. Pag 8. contained a total of 32,216 variants, of which 24,194 were LR germline variants (found in the exome of the blood cells). How many variants are generally found in a germline analysis? How many of them are SNPs?

The new table 2 lists in detail the different types of variants/polymorphisms we obtained in each tumor as well as in the blood of the liver recipient including those surrounding the exon that are also captured by the exonic probes. The results were updated with the most recent population data and the numbers are slightly different than what we reported from the time of our submission last year.

5. From tables 2 and 3, 55 rare variants are found. The authors should declare the meaning of "rare". We did define the meaning "rare" in the text as follow: "ExAC frequency ≥ 0.0001 and ≤ 0.0009 " and following reviewer 3 comment 3 we now elaborate on the fact that some of those can also be somatic but without the donor and other recipient germline, it is impossible to clearly assess their origin. However, none had any importance in the development of cancer.

6. Was BRAF-V600 mutation a germline or somatic mutation?

The pathology report we reviewed mentioned that the mutation was detected in the tumor by immunohistochemistry and our sequencing result indicates that it is a somatic mutation: its allelic fraction in the donor tumor is 0.25. A germline mutation would typically have an allelic fraction of 0.5.

7. What is the calculation to estimate tumor purity based o BRAF V600E?

The allelic fraction is first obtained with the ratio variant coverage over total coverage (reads with a mutation over the total number of reads). This ratio is then multiplied by two to calculate the purity, since there are two alleles in the genome. We used BRAF allelic fraction since BRAF is a predominant cancer driver and can be assumed to occur earlier in oncogenesis. We and others used a similar approach to estimate the pancreatic tumor purity based on the KRAS mutation allelic fraction (Nature, 491: 399-405, 2012).

8. According to the journal scope, this sentence "The datasets generated during the current study are available from the corresponding author on reasonable request" should be deleted and raw and downstream data uploaded on GEO or ENA database.

We have initiated the process to deposit the sequences in dbGAP.

Reviewer #2 (Comments to the Authors (Required)):

1. The authors tried to track the relationship of the donor and the recipient's tumors using next generation sequencing technology. NGS was used to profile whole genome and/or whole exom of the donor and the recipient's tumor and germlime. They identified 55 rare variants and 156 coding somatic mutations that were shared among the donors and the recipients' lesions. They also identified a chromosomal deletion spanning CDKN2A/B genes that were shared among the samples. The paper conclude that the donor/recipient cancer transmission was confirmed at a molecular level based on the similarity of the mutational profiles of the donor and recipient samples and the recipient tumor is a clonal expansion of the donor tumor.

Thank you to this reviewer for the summary of this study.

2. Transmission through transplantation of an organ with undetected metastasis has been known to occur as shown in various previous publications. So it is essential to show through mutation profiling that there are cancer cells or cancer causal mutation in the donor organs. In this paper, it is not clear whether the transmission was caused by the undetected tumor cells or causal mutations in the

transplanted organ. Is it possible to identify/separate donor organ somatic mutations from the recipient cancer tissue and establish the transmission path from the original donor's tumor to the donor's healthy transplant organ and finally to the recipient.

In this study we established the somatic profile of the metastasis of the liver recipient and traced back the same mutation profile in the donor tumor and in the tumor of the other organ recipients (see response to the first reviewer comment 1). Our study therefore shows that the transmission included mutations that were shared in different recipients, which is strong evidence that the mutations arose prior to transplantation – i.e. within the donor. If the tumors had instead each evolved from causal mutations after organ donation, as suggested by the reviewer, each tumor would have evolved with a different profile and the resulting somatic profile of each lesion would have been far different from one another.

The evidence we used to establish this included:

- The identification of the same mutation profile (187 mutations) in all tumors. The profile contained the same driver genes (such as *BRAF* and *PIK3CA*) but most importantly the same passenger gene mutations (such as 58 synonymous mutations);
- Y chromosomal variants and more than 22,000 donor polymorphisms identified in the omentum metastasis;

Our design and analysis are explained in detail in our answer to the first reviewer comment 1 and in the new paragraphs, figure, and tables we added.

3. The authors did great work in profiling and identifying the somatic mutations of the tumor samples and tracking the origin of the recipient's cancer, but failed to show how the donor's tumor genome sequencing information can be used to enhance organ transplant safety as the paper's title claimed.

As stated in our discussion, as we learn more about genetic correlates of tumor invasion and metastases, the methods demonstrated here will be even more powerful for the prevention of accidental transmission.

Reviewer #3 (Comments to the Authors (Required)):

The authors investigate a case study of cancer transmission through transplantation, focusing on the genetic profiles of 4 tumors: the donor's tumor sample, and 3 (of 4) recipient's tumor samples. The introduction describes the state of organ donation from donors with CNS tumors, and the overall low risk for transplantation of non-CNS organs from this type of donor. The anaplastic pleiomorphic xanthoastrocytoma (PXA) CNS tumor of the donor was not expected to metastasize outside of the CNS; nevertheless, aggressive neoplasms were detected in the recipients shortly after transplantation. They note previous studies of transmission of cancer through transplantation, where origin of the cancers is primarily verified by histology and analysis of markers found in the donor's tumor. The authors propose 2 questions to be answered in this study: (1) can the origin of the recipients' tumors be determined by NGS analysis? and (2) does this open a new perspective on using NGS methods to reduce the likelihood of transmission at transplantation? The first question is sufficiently addressed in the paper, but the second question (and the related title of the manuscript) is primarily speculation and related claims

should be toned down accordingly.

The results describe each of the patients and neoplasms in the study in great detail, including their clinical characteristics and histology/morphology/immunohistochemistry analysis. They then describe a NGS analysis of the donor and recipient tumors, which appears to be the first time this technology has been applied to investigate transplant-associated neoplasms. The NGS analysis of the tumors focuses primarily on comparing the omentum metastasis of the liver recipient patient to the blood sample of the liver recipient patient. This choice seems driven by the availability of samples (i.e. no germline sample from the donor or other recipients). By comparing this metastatic sample to the germline of that patient, the authors are able to identify variants unique to the cancerous cells in that patient, though they cannot technically distinguish rare variants from the donor's germline from somatic variants unique to the cancer cells. By filtering on population frequency, the authors define a "somatic set", demonstrate that these variants are present in the donor's tumor and the tumors of 2 other recipients, and highlight some of the cancer-associated genes mutated (BRAF, PIK3CA, SDHC, FANCD2, DDR2, NCAM1). They also identify a chromosome 9 deletion in the donor tumor through whole genome sequencing, and verify the presence of this deletion in the other tumor samples using MLPA. The authors briefly mention variants unique to each lesion or shared by a subset of lesions, but do not quantify or include details on these. Further investigation into the PIK3CA variant is shown, due to the discrepancy in its detection between the first and second donor samples. The authors argue that the higher PIK3CA frequency in the resected sample indicates aggressive growth of this clone in the 10 days between when the samples were taken. The authors provide sufficient evidence that NGS-based mutation profiles have the potential to confirm the origin of tumors that arise after transplantation. The authors discuss the mutations found in this tumor sample that are not typically associated with PXA, and claim that their association with other more aggressive neoplasms could provide an explanation for the transmission of cancer in this case.

I have the following concerns and suggestions that should be addressed to improve the manuscript:

1. Figure 1 and the related results section has several issues that need improvement. The legend reads like a results section and does not actually describe what is shown in the figure flowchart. The last 2 rows, "tissue sequenced" and "exome association" are confusing and possibly inaccurate. For example, there is mention in the figure of recipient blood for the kidney and pancreas patients, but it does not appear that was sequenced. It is generally difficult to parse from this figure and the text exactly what tissues were sequenced with what technologies. The text of the "mutational profile and comparative genomic analyses" is confusing overall and should be rewritten. For example, the authors state: "None of the lesions growing inside the transplanted organs could be used since they were infiltrated by the donor normal cells, hence the donor germline, which could result in false assumption", but then go on to "use" these samples in the analysis.

We simplified Figure 1 and added a new figure (now Figure 3) to illustrate the study design. We explained in greater detail the study design and illustrated in a table each tumor sequencing result (please also refer to our answer to the first reviewer comment 1).

2. The analysis focuses on the "somatic set" of variants identified in the omentum sample which is shared between the donor tumor and the recipient tumors. The authors should quantify in the text, and include details in a supplemental table, of all variants identified that were NOT shared between all

samples (i.e. the NF2 and NOTCH4 variants). Without this, the reader cannot assess the similarity of the profiles, since only the agreements are described in detail. The lack of clarity on their precise origin does not preclude the authors from describing what they do know about these variants.

A new summarized table was added to the text as well as a supplementary table S4 listing the variants shared between 2 and 3 tumors as well as unique to each tumor. We redid the analysis using the most recent population data in all exomes. In this new analysis, the NOTCH4 variant was listed as a population polymorphism and was removed from the list of somatic variants. It is possible that other polymorphisms remained in the tables but they cannot be identified without the germline exome of each patient.

3. The use of ExAC frequency to distinguish between "rare variants" and "coding somatic mutations" is a reasonable step but is not conclusive. True somatic variants could have frequencies in the range indicated for rare variants, and rare variants could be absent from the ExAC database. It would be more accurate to call this set "likely somatic".

We are now treating these variants as rare polymorphisms and potentially somatic variants and explain that it is impossible without the donor and recipient germline to exactly determine what they are.

4. This study would have been more straightforward if a germline sample from the donor was available. In that case, somatic variants could have been easily identified and evaluated in all other samples. The authors should address this limitation in the study design.

We totally agree with the reviewer that having a normal sample from the donor as well as from the other recipients would have greatly facilitated the identification of each somatic profile and the comparison between them. We added several paragraphs in the Result section as well as a new Figure (Fig 3) to explain the restriction we were facing and the alternative approach we took, an approach that was nevertheless conclusive of the origin of the tumors.

5. To strengthen the analysis and evidence of the similarity of the mutation profiles between tumors, the authors could to show a global comparison of allele frequencies (i.e. a scatter plot of the allele frequency columns in the supplemental tables).

The fact that the tumors do not have the same purity (Donor 47%; LR-Om 97%; KR 78%; and KPR 78%) and that they have not been preserved the same way, can affect the allelic fraction of the mutation in each tumor. We are inserting here the scatter plots for the reviewer to see. We might have misunderstood the reviewer request but presently we do not see how these plots strengthen the analysis and evidence of the similarity of the mutation profiles between tumors.

6. The conclusion that the higher PIK3CA frequency in the resected sample compared to the fragment sample indicates aggressive growth of this clone in the 10 days between when these samples were taken does not consider alternative explanations. Spatial heterogeneity of the PIK3CA subclone and sampling bias of the two samples could also explain the discrepancy. The authors use this evidence to support their claim that repeated NGS sampling and evolution of a mutation profile could indicate aggressiveness and potential spread of the tumor, but this is mostly unfounded speculation based on this one data point. The authors should discuss alternative explanations for this finding.

There is indeed spatial heterogeneity in a tumor. As suggested, we added this alternative explanation. However, we also mentioned that only the cells that had a PIK3CA mutation crossed the brain barrier. In 2 out of 3 lesions, the PIK3CA had an allelic fraction equivalent to BRAF (0.45 and 0.49, 0.39 and 0.39). This supports the importance of this driver gene mutation in the propensity to invade and disseminate.

7. Linking several mutations found in this PXA case with an aggressive phenotype based purely on the presence of these gene mutations in other malignancies is not well supported by the literature and the current understanding in the field. Using these mutations as a convenient explanation for the aggressiveness of this case and suggesting that this same prediction could be made in future cases without knowing the outcome is a little misleading. The authors suggest that genetic profiling will be useful as "we learn more about correlation between comprehensive mutation profile and invasion causality", which is likely true, but I find their conclusions specific to this case to be overstated based on current knowledge.

The donor suffered an early regrowth and died of a disease that is normally curable. His tumoral cells travelled through the brain barrier, sustained the blood pressure, invaded and survived in multiple organs in a cellular environment totally different than the brain, and expanded furiously in a manner of few weeks in the recipients. These cells had all the same genetic profile. Without relying on any publications, such cells and their genetic profile can be defined as aggressive. We realized that our opinion might not have been properly expressed in the manuscript, so we modified our text. We never intended to suggest that other cases will have the same mutations, neither to put any blame on the consequence of this transplant. However, we want this study to emphasize that sequencing studies should be done to learn more about transmission by transplantation.

8. Minor figure suggestions: the Figure 3 axis labels and text is small and difficult to read. The legend does not describe what is actually plotted (what do the colors represent, what are the error bars, etc.). Table 2 should indicate which samples this list of events are from.

The size of the graphs and the size of the axis labels in Figure 3 (now Figure 4) were increased. The figure legend was modified to include a better description of the figure components.

Table 2 title has been modified.

9. In the authors' discussion of the potential for NGS to assist in transplant decisions in the future, they could consider discussing detection of circulating tumor cells or cell-free tumor DNA as a means of assessing potential metastasis.

As suggested, we added this subject to the discussion.

10. Sequencing data should be deposited in a repository (dbGAP, EGA) as is standard in the field for exome and whole genome sequencing.

We have initiated the process to deposit the sequences in dbGAP.

June 1, 2021

Re: Life Science Alliance manuscript #LSA-2020-00941-TR

Dr. Marie-Claude Gingras
Baylor College of Medicine
Human Genome Sequencing Center
One Baylor Plaza, MSC226
Houston, TX 77030-3411

Dear Dr. Gingras,

Thank you for submitting your revised manuscript entitled "Sequencing of a central nervous system tumor demonstrates cancer transmission in an organ transplant" to Life Science Alliance. The manuscript has been seen by the original reviewers whose comments are appended below. While the reviewers continue to be overall positive about the work in terms of its suitability for Life Science Alliance (LSA), some important issues remain.

As you will note from the reviewers' comments below, the reviewers are mostly happy with the revised manuscript. Both Rev 1 and 2 still seem to have an outstanding concern about the discrimination between germline and somatic mutations. We would, thus, like to invite you to submit a revised version that addresses this remaining concern of the reviewers.

Our general policy is that papers are considered through only one revision cycle; however, given that the suggested changes are relatively minor, we are open to one additional short round of revision.

Please submit the final revision within one month, along with a letter that includes a point by point response to the remaining reviewer comments.

B. MANUSCRIPT ORGANIZATION AND FORMATTING:

Sincerely,

Shachi Bhatt, Ph.D.
Executive Editor
Life Science Alliance
<http://www.lsajournal.org>
Tweet @SciBhatt @LSAJournal

Reviewer #1 (Comments to the Authors (Required)):

The authors performed a complete revision of the manuscript, with a great effort in clarify important points as requested. This reviewer believes that the authors performed a valuable, interdisciplinary investigation regarding the emerging topic of donor/recipient cancer transmission after organ transplantation. The authors broadly ameliorated the description of investigation after the first round of revision however, this reviewer still finds a lack of data and, unfortunately sample availability to support the demonstration of the cancer transmission after in a transplanted organ.

I consider the manuscript still valuable for the scientific community, and although the experimental design is not anymore ameliorable due to the lack of sample availability, data need to be further processed to result fully convincing.

In particular, this reviewer still finds it challenging to interpret the enormous amount of variants found in germline and tumor samples that go beyond the number of 100k in each sample. In particular, it is not easy to explain more than 100k variants assigned to non-coding regions while the experiments are Whole Exome Sequencing. I suggest the authors to re-analyze the data using another variant caller such as GATK4 following the best practices from the Broad Institute and discriminate the analysis between germline and somatic according to the best practice. Overall, the authors need to exclude any possibility that the called variants are artifacts.

Reviewer #2 (Comments to the Authors (Required)):

1. This is a revised version of a submission a year ago. The authors had done a great work in revising the paper, adding more discussions on the results, adding details and data tables

requested by previous reviewers. Most of my concerns were addressed properly.

2. The newly added sup tables are informative.

3. The only concern I have now is: When reviewing table S2 and S3, it appears when you plot the donor variants allelic frequency distribution, there is actually a peak around 0.5, this may indicate that in table S2 and S3, many of the donor somatic mutations are actually germline mutations. The authors clearly had done the filtering step thoroughly, there should not be many germline mutations left in the final table.

Reviewer #3 (Comments to the Authors (Required)):

The authors have greatly improved the clarity of the manuscript and sufficiently addressed my concerns. I have no further suggestions.

Reviewer #1

This reviewer still finds it challenging to interpret the enormous amount of variants found in germline and tumor samples that go beyond the number of 100k in each sample. In particular, it is not easy to explain more than 100k variants assigned to non-coding regions while the experiments are Whole Exon Sequencing. I suggest the authors to re-analyze the data using another variant caller such as GATK4 following the best practices from the Broad Institute and discriminate the analysis between germline and somatic according to the best practice. Overall, the authors need to exclude any possibility that the called variants are artifacts.

The reviewer might not be aware that DNA oligonucleotide capture reagents, including the Nimblegen probes developed here at the Human Genome sequencing Center (HGSC) to capture exonic DNA, expand to target 2 million bases outside the exons. These probes also capture DNA that expands further into the introns and in the 5' and 3' flanks of the targeted genes. The capture also contains a considerable amount of other off-target DNA at sufficient coverage to support variant calls. Regions outside the CCDS have an overall higher rate of variation than the coding region (see Bainbridge *et al* 'Targeted enrichment beyond the consensus coding DNA sequence exome reveals exons with higher variant densities' Genome Biol. 2011 12:R68, among others, for discussion). In the majority of genetic studies, where the focus is on curated coding variation, non-exonic data are usually not mentioned and are put aside in the first stage of the analysis. In cancer studies, where we have been analyzing WES data for more than a decade in programs including the The Cancer Genome Atlas Research Network, these ranges are much more familiar. Several papers using similar variant identification and analysis pipelines have been published in Nature and other high ranked Journals.

Further, the initial variant calls in this and other similar studies are performed at relatively low stringency in order to minimize false negative rates. Those familiar with the current type of study are also familiar with this level of variation seen in similar analyses. Here, we point to the amount of variation detected in the coding regions of the DNA sample from the blood of the liver recipient as a relevant metric: we detected 23,282 variants. In other exome sequencing experiments from blood DNA samples we routinely see 23,000 and so the false positive rate is not too high.

The suggestion to switch to the alternate variant caller (GATK) is not appropriate. The cited 'best practices' should not be conceptually confused with the more familiar use of 'best practice' in medical care that is rooted in established clinical practices. Here, in the context of variant calling software, the use of 'best practices' is simply nomenclature. We have extensive experience, including a nearly 20 year history, in the design and calibration of variant calling software. If we were to apply GATK to this study there would be nearly identical results and the differences would not be germane to the conclusions. If we were to have approached the variant calling differently we would have (i) possibly utilized different stringencies to result in an initially narrower call set and alleviate the concerns of others like reviewer one, or (ii) if we believed that the ultimate performance of the variant calling software was at issue, defaulted to the use of Deep Variant, which is superior to GATK and indeed, our own software, in every way except for the cost of the analyses. However, as pointed out below, these are moot issues – the logic that was demonstrated in this study was not affected by the possibility of false positive variant calls in either the initial determinations made in the liver donor blood and the omentum samples or in later analyses.

We acknowledge that the logic flow in this study is hard to follow - and therefore has been difficult to present. The reviewer initially asked for a presentation of more of the data from the analyses and hence we presented Table 2 (inserted below for his/her convenience). We ask that these additional data be considered with the understanding that the population analysis and comparison largely ameliorates any of the kind of concerns that are expressed. Simply put, if a fraction of the variants were artifacts (false positives), there would be a decreased likelihood that they would not be found in the normal population. As an example, in the case of the donor, we identified 121,712 non exonic variants of which 118,933 could be found in population databases. The same can be said for the KR and KPR tumors.

Variant classification	Liver Recipient					Donor			Kidney Recipient			Kidney Pancreas Recipient		
	Omentum metastasis	Blood	Blood germline removed	Common SNPs	Potential somatic variants	Lesion	Common SNPs	Potential somatic variants	Lesion	Common SNPs	Potential somatic variants	Lesion	Common SNPs	Potential somatic variants
UTR, unspecified exonic	9,741	8,014	1,727	1,598	129	8,674	8,497	177	10,816	10,567	249	11,511	11,212	299
Indel	404	302	102	92	10	333	304	29	414	367	47	446	388	58
Nonsynonymous (missense)	14,173	10,933	3,240	3068	172	9,860	9,699	161	13,797	13,470	327	14,451	14,120	331
Nonsense	115	84	31	25	6	66	63	3	103	91	12	98	89	9
Synonymous (silent)	15,525	11,866	3,659	3544	115	11,269	11,178	91	15,658	15,457	201	16,497	16,296	201
Splice Site	130	97	33	29	4	112	107	5	144	135	9	134	131	3
Non exonic, ncRNA	118,464	100,074	18,390	16,924	1,466	121,712	118,933	2,779	145,911	142,632	3,279	153,175	149,074	4,101
Total variant count	158,552	131,370	27,182	25,280	1,902	152,026	148,781	3,245	186,843	182,719	4,124	196,312	191,310	5,002

Reviewer #2

When reviewing tables S2 and S3, it appears when you plot the donor variants allelic frequency distribution, there is actually a peak around 0.5. This may indicate that in table S2 and S3, many of the donor somatic mutations are actually germline mutations. The authors clearly had done the filtering step thoroughly, there should not be many germline mutations left in the final table.

Table S2 contains rare variants found in the population. It is expected that their allelic fraction in the donor tumor reflect their probable germline origin. Reviewer #3 had asked in the first revision to label those potentially somatic variants. We are now listing them as probable rare or specific donor polymorphisms.

Table S3. Without the donor germline exome, we relied on population databases to identify germline variants. These databases will not contain variants specific to an individual and we could not differentiate the rare and specific variants from the somatic variants using these databases. So we used a cut off of ≤ 0.00009 to split the variants into 2 tables: probable rare and specific donor variants (Table S2) and potential somatic variants (Table S3) and we did not consider the variant allelic fraction. We understand the reviewer concern, and we removed some variants from Table S3 and moved them in Table S2. We also modified the title of both tables and specify their content in the text:

“We observed 137 coding region variants with an ExAC frequency ≤ 0.000094 (probably somatic) and 68 variants with an ExAC frequency ≥ 0.0001 and ≤ 0.0009 and 33 variants with an allelic fraction of 0.49-0.51 in the donor tumor (probably rare or specific donor germline variants) shared among all tumors...”

Table 3 in the manuscript was also modified to reflect these changes.

July 6, 2021

RE: Life Science Alliance Manuscript #LSA-2020-00941-TRR

Dr. Marie-Claude Gingras
Baylor College of Medicine
Human Genome Sequencing Center
One Baylor Plaza, MSC226
Houston, TX 77030-3411

Dear Dr. Gingras,

Thank you for submitting your revised manuscript entitled "Sequencing of a central nervous system tumor demonstrates cancer transmission in an organ transplant". We would be happy to publish your paper in Life Science Alliance pending final revisions necessary to meet our formatting guidelines.

- please upload your main and supplementary figures as single files
- please use the [10 author names, et al.] format in your references (i.e. limit the author names to the first 10)
- if possible please provide one figure per file
- please integrate the Supplementary Methods and associated References into the main Materials & Methods and References sections. We do not have a size limit on these sections.
- please add scale bars to Figure 2, and indicate the scale bar sizes in the Legend

A. FINAL FILES:

B. MANUSCRIPT ORGANIZATION AND FORMATTING:

Sincerely,

July 12, 2021

RE: Life Science Alliance Manuscript #LSA-2020-00941-TRRR

Dr. Marie-Claude Gingras
Baylor College of Medicine
Human Genome Sequencing Center
One Baylor Plaza, MSC226
Houston, TX 77030-3411

Dear Dr. Gingras,

Thank you for submitting your Research Article entitled "Sequencing of a central nervous system tumor demonstrates cancer transmission in an organ transplant". It is a pleasure to let you know that your manuscript is now accepted for publication in Life Science Alliance. Congratulations on this interesting work.

DISTRIBUTION OF MATERIALS:

Again, congratulations on a very nice paper. I hope you found the review process to be constructive and are pleased with how the manuscript was handled editorially. We look forward to future exciting submissions from your lab.

Sincerely,
